# A *Candida parapsilosis* Overexpression Collection Reveals Genes Required for Pathogenesis

**DOI:** 10.3390/jof7020097

**Published:** 2021-01-29

**Authors:** Sára E. Pál, Renáta Tóth, Joshua D. Nosanchuk, Csaba Vágvölgyi, Tibor Németh, Attila Gácser

**Affiliations:** 1Department of Microbiology, University of Szeged, Közép Fasor, 6726 Szeged, Hungary; palsara0713@gmail.com (S.E.P.); renatatth@gmail.com (R.T.); csaba@bio.u-szeged.hu (C.V.); narvaltm@gmail.com (T.N.); 2Departments of Medicine and Microbiology and Immunology, Albert Einstein College of Medicine, New York, NY 10461, USA; josh.nosanchuk@einsteinmed.org; 3MTA-SZTE Lendület Mycobiome Research Group, University of Szeged, 6726 Szeged, Hungary

**Keywords:** *Candida parapsilosis*, gene overexpression, mutant collection, virulence factors

## Abstract

Relative to the vast data regarding the virulence mechanisms of *Candida albicans*, there is limited knowledge on the emerging opportunistic human pathogen *Candida parapsilosis.* The aim of this study was to generate and characterize an overexpression mutant collection to identify and explore virulence factors in *C. parapsilosis*. With the obtained mutants, we investigated stress tolerance, morphology switch, biofilm formation, phagocytosis, and in vivo virulence in *Galleria mellonella* larvae and mouse models. In order to evaluate the results, we compared the data from the *C. parapsilosis* overexpression collection analysis to the results derived from previous deletion mutant library characterizations. Of the 37 overexpression *C. parapsilosis* mutants, we identified eight with altered phenotypes compared to the controls. This work is the first report to identify *CPAR2_107240*, *CPAR2_108840*, *CPAR2_302400*, *CPAR2_406400*, and *CPAR2_602820* as contributors to *C. parapsilosis* virulence by regulating functions associated with host-pathogen interactions and biofilm formation. Our findings also confirmed the role of *CPAR2_109520*, *CPAR2_200040,* and *CPAR2_500180* in pathogenesis. This study was the first attempt to use an overexpression strategy to systematically assess gene function in *C. parapsilosis*, and our results demonstrate that this approach is effective for such investigations.

## 1. Introduction

*Candida* species are responsible for the vast majority of systemic fungal nosocomial infection cases. Numerous studies from different parts of the world report increasing rates of systemic candidiasis [1,2,3,4]. Non-*albicans Candida* (NAC) species such as *C. glabrata*, *C. parapsilosis*, *C. krusei*, *C. auris*, and *C. tropicalis* have significantly contributed to this continuous rise in the number of candidiasis cases, which highlights the importance of these emerging species [5,6,7]. Remarkable changes in medical practice that subvert host effector responses, such as the robust use of immunosuppressive therapies, can also affect the epidemiology of a pathogenic species and increase the incidence of fungal infections [8,9,10].

There is an enormous amount of data available in the literature on the virulence mechanisms of *C. albicans*, while our knowledge lags behind for emerging NAC opportunistic human pathogens, such as *C. parapsilosis* [11]. Although *C. parapsilosis* is a commensal yeast of the normal human skin, it can, in contrast to the strict association of *C. albicans* with humans, also survive in diverse environments. *C. parapsilosis’* increasing role in low birth weight neonatal infections highlights the importance of performing extensive surveys of its physiology and virulence factors [12,13,14,15]. *C. parapsilosis* is able to form pseudohypha, but not true hypha, and its elevated tolerance to echinocandins are additional specific attributions with clinical relevance [16,17]. So far, several groups have explored specific gene functions in *C. parapsilosis* [18,19,20]. The first genome sequence of a *C. parapsilosis* strain (CDC317) was published in 2009 [21]. Later, three more strains derived from human and environmental sources were also sequenced and comprehensively analyzed, enabling further investigations of large scale targeted genomic alterations in this species [21,22]. These and other studies also revealed genomic and phenotypic alterations between *C. albicans* and *C. parapsilosis*, suggesting that these two pathogenic species have evolved different strategies to invade the host [23,24]. According to the Candida Genome Database (CGD) only 1.35% of the protein coding genes of *C. parapsilosis* have been verified to date, while this number is 27.9% in the case of *C. albicans* [25,26]. Taking into account the unique and distinctive interspecies characteristics of *C. parapsilosis* in the *Candida* clade, further investigation is urgently needed to reveal factors associated with the pathobiological mechanisms of the species.

The conventional method to reveal gene function is the deletion of a given gene and observation of the phenotype of the knock out (KO) mutant. Comprehensive KO mutant libraries and their characterizations are accessible for many organisms, including *C. parapsilosis* [27,28,29,30]. Pathogenesis associated gene functions such as biofilm formation, phenotypic profile, and transition or genes responsible for drug resistance or transcriptional regulator networks have also been identified in both *C. albicans* and *C. parapsilosis* using KO methods. Even in silico gene deletion for drug development has been carried out with *C. albicans* [31]. This approach was aided by the rapid evolution of the molecular techniques replacing the laborious *CaSAT1* flipper, or the double auxotrophy complementation approaches with CRISPR/Cas9 system that simplified the generation of loss of function mutations [18,28,32,33,34,35]. However, in some cases, the resulting phenotype achieved by gene deletion has been uninformative, or the KO approach has failed, especially when essential genes are in question. Furthermore, in diploid organisms deletion of both alleles for the targeted gene is required. The artificial overexpression (OE) of a gene can circumvent the aforementioned drawbacks of the KO method. In addition to this, an OE strategy can be utilized to map regulatory rate-limiting steps in gene interaction studies [36]. Systematic library analysis as well as specific gene examination studies have been published using OE mutant strains in many organisms [37,38,39,40,41]. Genes involved in biological processes associated with fungal pathogenesis, such as biofilm formation, invasive hyphal growth or drug resistance, have also been successfully identified using this approach in *C. albicans* and other *Candida* species [42,43,44]. Moreover, the OE approach can also complement the KO library screens to explore and validate the function of a gene [36,45,46,47,48].

Although extensive studies in gene OE in *Saccharomyces cerevisiae* and *C. albicans* have enriched our knowledge on their fitness and virulence [49,50,51,52], no such an effort has yet been achieved in the case of the neonatal pathogen *C. parapsilosis*. Therefore, we have generated and characterized an OE collection in *C. parapsilosis* using a recently adapted system by Németh and co-workers [53] and explored the collection in order to reveal novel virulence traits.

## 2. Materials and Methods

### 2.1. Strains and Growth Conditions

*C. parapsilosis* and *Escherichia coli* strains used in this study are listed in Appendix A. Yeast strains were stored at −80 °C in YPD (0.5% (*m*/*v*) yeast extract, 1% (*m*/*v*) peptone, and 1% (*m*/*v*) D-glucose) media supplemented with 20% (*v*/*v*) glycerol. Yeast strains were routinely maintained on YPD plates supplemented with 2% (*m*/*v*) agarose and 100 unit/mL Penicillin-Streptomycin (PS) solution and maintained at 4 °C. For the experiments, fungal strains were grown in YPD media supplemented with 100 unit/mL PS in an orbital shaker at 30 °C overnight, then 1 µL of the overnight culture was transferred into 3 mL of fresh YPD media and cultivated under the same conditions overnight. Two milliliters of the suspension was collected and washed twice with sterile 1× PBS, diluted then counted with a Burker-chamber. Specific cell concentrations were set using 1× PBS.

Prototroph transformants were selected and maintained on solid minimal media, containing YNB (0.19% *m*/*v* Yeast Nitrogen Base without amino acids) media supplemented with 2% (*m*/*v*) glucose, 2% (*m*/*v*) agar, 100 unit/mL PS, and 10% (*v*/*v*) 10× Dropout (DO) amino acid solution (without leucine) [28].

*E. coli* strains were grown on LB plates (1% (*m*/*v*) NaCl, 1% (*m*/*v*) tryptone, 0.5% (*m*/*v*) yeast extract, and 2% (*m*/*v*) agar) supplemented with 10 µg/mL tetracycline for strain 2T1. Selection of colonies bearing the plasmids was carried out on LB plates supplemented with 50 µg/mL kanamycin A or 100 µg/mL ampicillin. Plasmid extraction from 2T1 or DB3.1 *E. coli* strains was performed by cultivating transformants in 2 mL LB liquid media in an orbital shaker at 37 °C with the appropriate antibiotics overnight.

Dulbecco’s modified eagle’s medium (DMEM) + 10% (*v*/*v*) heat-inactivated fetal bovine serum (HI FBS) + 100 unit/mL PS media was used for pseudohypha detection experiments and for the maintenance of the J774.2 murine cell line.

### 2.2. Construction of the Overexpression Cassettes

Overexpression constructs were generated using Gateway^TM^ cloning method with pDONR^TM^221 for BP and pDCpOE-L-N5L for LR cloning [53]. Target ORFs were amplified with an attB1 site at the 5′ end, and an attB2 site at the 3′ end. Additionally, the 3′ end also contained a 20 nt long unique bar code sequence to enable rapid identification by molecular methods (Appendix A). BP and LR cloning were performed according to the manufacturer’s instructions. *E. coli* strain 2T1 was transformed with BP and LR cloning products and used to propagate pENTRY and pECpOE, respectively, while pDONR^TM^221 and pDCpOE-L-N5L plasmids were propagated in strain DB3.1.

### 2.3. Generation of the Overexpression Mutants

The pECpOE-ORF-L-N5L plasmids were linearized by *Stu*I digestion overnight, precipitated, washed with 70% (*v*/*v*) ethanol, air-dried, and suspended in 10 µL distilled water. Three micrograms of the linearized plasmid was used to transform the leucine auxotroph derivative of *C. parapsilosis* (CPL2) using the ssDNA/LiAc/PEG mediated heat-shock method as described by Holland and co-workers [28]. Cells were plated onto minimal media and incubated for three days at 30 °C.

### 2.4. Validation of the Mutant Strains

To verify the proper integration of the cassettes, we first performed rapid DNA isolation followed by colony PCR involving primers specific to the given ORF and downstream from the site of the integration [28]. See primer sequences in Appendix A. Further verification was managed using Southern blot. Briefly, 10 µg of total genomic DNA was digested with *Eco*RI overnight. A digoxigenin labeled CpN5L downstream probe was generated by amplifying the given region from *C. parapsilosis* CLIB214 genomic DNA with CpN5LDo1SouFor and CpN5LDo2SouRev primers using the DIG DNA Labeling Kit (Roche) according to the manufacturer’s instructions (Appendix A). For real-time PCR analysis, extraction and reverse transcription of the RNA to cDNA was performed using Ambion, Ribopure^TM^-Yeast RNA Isolation Kit, and the RevertAid First Strand cDNA Synthesis Kit (ThermoFisher), respectively, according to the instructions of the manufacturers. Real-time qPCR was carried out as previously described [29] in a Bio-Rad Real-Time PCR detection system. Data were normalized to CLIB214 wild type strain, and relative transcription levels were determined using the housekeeping gene *CpTUB4* (*CPAR2_500510*) as an internal control. The RNA extraction was performed from the pool of three independent cultures mixed in an even ratio. To evaluate the results, the 2^−∆∆Cq^ comparison method was used with Bio-Rad CFX Manager software. Primers for these experiments are listed in Appendix A.

### 2.5. Characterization of the OE Mutant Strains

#### 2.5.1. Growth Kinetics Measurements in Liquid Cultures

The experiments were performed three times except for those mutants that showed no difference compared to the control in the first two experiments. Three different control *C. parapsilosis* strains-the wild type strain (CLIB214), the reintegrated strain of the double auxotroph strain (CPRI), and the mCherry^OE^ mutant-were used in growth kinetics, stress tolerance, and antifungal susceptibility experiments.

To monitor the growth capabilities of the generated mutants, kinetic studies were performed in YPD + 100 unit/mL PS and YNB (0.67% Yeast Nitrogen Base without amino acids) + 0.5% glucose + 100 unit/mL PS liquid medium under static conditions at 30 °C for 24 h, and the OD_600_ was determined every hour after one-minute of shaking (200 rpm) (SpectroStar Nano). Tests were performed three times in triplicates in 48-well plates. Initial concentrations were determined at 5 × 10^5^ cells/mL, where 100 µL of the suspension was added to each well loaded with 900 µL liquid medium.

#### 2.5.2. Viability Tests under Different Stress Conditions on Solid Media

Viability of the strains was tested under the stress conditions listed in Appendix A. Four step ten-fold dilutions were prepared with 10^5^ to 10^2^ cells per 5 µL. Growth was examined at 30 and 37 °C after 48 h of incubation. Heat dependency was examined by incubating the cells at 20, 25, 30, 37, and 40 °C on YPD solid medium. In addition, growth was investigated on solid minimal media (0.19% YNB + 0.5% glucose + 100 unit/mL PS) with or without the addition of 10% (*v*/*v*) HI FBS at 30 and 37 °C.

#### 2.5.3. Morphology Switch and Pseudohypha Forming Capacity

The morphology of OE strains was examined under a light microscope. Prior to examination, each strain was grown in liquid YPD + 100 unit/mL PS media, collected, washed three times, suspended in 1× PBS solution, and pipetted onto glass slides.

The pseudohypha forming capacity was investigated after growing the cells in liquid YPD + 100 unit/mL PS and DMEM + 10% (*v*/*v*) HI FBS + 100 unit/mL PS media at 37 °C, 5% (*v*/*v*) CO_2_ for 24–48 h. After one day of incubation, the samples were examined by light microscopy. After two days, each sample was measured with Amnis^®^ FlowSight^®^ flow cytometer. Data were analyzed by the IDEAS Software (Amnis).

#### 2.5.4. Biofilm Assay

Biofilm inducing conditions were 0.67% (*m*/*v*) YNB + 0.5% (*m*/*v*) glucose + 100 unit/mL PS liquid media incubated at 37 °C in the presence of 5% (*v*/*v*) CO_2_ for 48 h without shaking. Experiments were carried out in 10% FBS precoated 96-well plates (polystyrene). After two washing steps, 20–20 µL 5 mg/mL 3-(4,5-dimethylthiazol-2-yl)-2,5-diphenyltetrazolium bromide (MTT) solution was added to the suspensions and incubated for 5 h under the same conditions followed by two washing steps with 1× PBS. Insoluble (formazan) residue was formed, which was dissolved in DMSO for 10–15 min, and OD_540_ were detected by spectrophotometry (SpectroStar Nano). Cell-free culturing media with MTT solution and MTT-free cell suspension samples were applied as controls. At least eight parallels were used in at least three experiments.

#### 2.5.5. Antifungal Susceptibility

Strains were prepared as described above then the concentration was adjusted to 3 × 10^4^ cell/mL by suspending the cells in Roswell Park Memorial Institute 1640 medium supplemented with 34.53 g/L 4-Morpholinepropanesulfonic acid (RPMI-MOPS). Two-fold dilutions of anidulafungin, caspofungin, and micafungin were prepared. Final drug concentrations were from 8 µg/mL to 0.0156 µg/mL with 3 × 10^3^ fungal cells in 200 µL final volume of RPMI-MOPS. Cells were incubated at 30 °C without shaking and monitored after 24 and 48 h. Minimum inhibitory concentration (MIC) values were determined according to the M27-A3 protocol, and MIC values were defined by the Appendix A [54,55]. MIC values for the applied drugs were defined as the lowest concentrations that resulted in at least 50% growth reduction. Experiments were performed in triplicates. In addition to mCherry^OE^, CPRI, and CLIB214 *C. parapsilosis* strains, a *C. krusei* strain with known antifungal susceptibility was also included as a control.

#### 2.5.6. Phagocytosis Assay

The murine macrophage-like cell line J774.2 and each mutant strain were grown as previously described. The fungi were stained with Alexa Fluor^TM^ 488 succinimidyl ester dye (Invitrogen) prior to the experiments, according to Papp and colleagues [17]. We then infected J774.2 macrophages with stained yeast cells using the multiplicity of infection (MOI) of 5:1 (pathogen:host) and co-incubated the cells at 37 °C, 5% (*v*/*v*) CO_2_ for 1 h. The wild type strain and non-infected J774.2 macrophages were used as controls to exclude autofluorescence. Samples were measured with Amnis^®^ FlowSight^®^ flow cytometer.

#### 2.5.7. In Vivo *Galleria mellonella* Infection Model

The larvae (BioSystems Technology-TruLarv^TM,^ Exeter, Devon, England) were kept at 4 °C and transferred to 30 °C one day before the experiment. Mutant yeast strains were tested at least two times with 20 larvae per sample. Concentrations of OE strains were adjusted to 5 × 10^7^ cell/mL. Each larva was injected with 10 µL suspension containing 5 × 10^5^ cells and incubated at 30 °C in dark. Survival was monitored daily for 10 days. Larvae injected with wild type strain, non-injected (witness) *G. mellonella* larvae, and 1× PBS injected larvae were used as controls.

#### 2.5.8. Cell Wall Composition Assay

To investigate putative alterations in certain cell wall components of the selected mutants (CPAR2_107240^OE^, CPAR2_108840^OE^, CPAR2_109520^OE^, CPAR2_200040^OE^, CPAR2_302400^OE^, CPAR2_406400^OE^, CPAR2_500180^OE^, and CPAR2_602820^OE^), ConA-FITC (Sigma-Aldrich, St. Louis, MO, USA), Calcofluor white (Sigma-Aldrich, St. Louis, MO, USA), and WGA-FITC (Sigma-Aldrich, St. Louis, MO, USA) were applied. Samples were prepared as previously described by Tóth and colleagues [29]. The first fluorescent dye mix contained 4 μL 2.5 mg/mL ConA-FITC and 0.5 μL 1 mg/mL Calcofluor white while the second contained 0.25 μL 2 mg/mL WGA-FITC in the final 100 µL volume of 1% BSA + 1× PBS solution. For detection, we used a Zeiss Axio Observer fluorescent microscope, and the means of the different fluorescent dyes were recorded with an Amnis^®^ FlowSight^®^ flow cytometer.

#### 2.5.9. In Vivo Mouse Infection Model

Selected mutant and wild type strains were prepared as described above, and cell concentrations were set to 2 × 10^8^/mL. *Mus musculus* BALB/c 7–8-week-old females (BRC, Szeged, Hungary, XVI./2015) were infected with 100 µL suspensions through their lateral tail veins. Animals were monitored daily and sacrificed after three days. Brain, liver, spleen, and kidneys were isolated, weighed, and homogenized under sterile conditions in 1x sterile PBS. Following subsequent dilutions, 50–50 µL of samples were plated on YPD plates and incubated for two days at 30 °C. Colony-forming units (CFUs) were determined, and the results were calculated and expressed in CFU/g tissue units. Four or five parallel animals were tested in two independent experiments.

### 2.6. Ethics Statement

Animal experiments were performed according to Hungarian national animal ethics guidelines (guideline 1998, XXVIII; 40/2013) and European animal ethics guidelines (guideline 2010/63/EU). The procedures were licensed by the Animal Experimentation and Ethics Committee of the Biological Research Centre of the Hungarian Academy of Sciences and the Hungarian National Animal Experimentation and Ethics Board (clearance number XVI./03521/2011), with the University of Szeged granting permission XII./00455/2011 and XVI./3646/2016 to work with mice.

### 2.7. Statistical Analysis

Statistical analysis was performed using GraphPad Prism 7 software. For the biofilm assay and the phagocytosis analyses, we used one-way ANOVA, Dunnett’s multiple comparisons test. To evaluate the results of the *G. mellonella* infection experiments Log-rank (Mantel–Cox) tests and, for the assessment of fungal infection in mice, Mann–Whitney tests were used. Statistically significant differences were considered at *p*-values of ≤ 0.05 (* *p* ≤ 0.05; ** *p* ≤ 0.01; *** *p* ≤ 0.001; **** *p* ≤ 0.0001).

## 3. Results

### 3.1. Generation of an Overexpression Mutant Collection in C. parapsilosis

The examined ORFs were preselected according to our preliminary experiments and related data from the literature (Appendix A). Transcriptional analysis of fungi co-incubated with THP-1 human monocytes was performed as previously described by Tóth and co-workers in order to identify genes related to pathogenesis based on their altered expression patterns during infection [29]. We selected 18 differentially expressed genes and 19 additional genes that also have known virulence-related ortholog functions in *C. albicans* or in *S. cerevisiae* based on literature prior to generating the OE mutant collection. The 18 differentially expressed genes were also tested in previous KO mutant library analyses in *C. parapsilsosis* [19,28,29].

To generate overexpression mutants, the Gateway^TM^ system was used. Our work is based on the previous work of Chauvel and colleagues with *C. albicans* [50], which we modified and optimized for *C. parapsilosis* [53]. Selected ORFs were amplified along with a 20 nt long bar code (See Appendix A TAG sequences) in the non-transcribed region to enable subsequent identification by molecular methods. Within the expression vector, the cloned ORFs were under the regulation of the TDH3 promoter of *C. albicans* (Appendix A). The plasmid was then linearized with *Stu*I digestion. The linearized plasmid (or integrative cassette) was then integrated into the CpNEUT5L (CpN5L) intergenic region of *C. parapsilosis* (Appendix A). We targeted the CpN5L region because, in contrast to the ortholog of *RPS1*, this locus not only provides a strong homogenous expression for exogenous constructs, but it does not affect the fitness of the transformed/modified strain [53,56]. The expression vector utilizes the *C. maltosa LEU2* marker for selection that is compatible with the leucine auxotroph laboratory strain of *C. parapsilosis* CPL2 [28]. A total of 37 ORFs were amplified, BP cloned, and then sequenced to ensure that there were no SNPs (single-nucleotide polymorphism) present causing any amino acid alterations. The correct integration of the linearized plasmids was verified by colony PCR and Southern-blot analysis (Appendix A). The relative expression of each gene was measured using qPCR and compared to CLIB214. We found that the expression levels varied across a wide range with increases from 2.6-fold to 675.7-fold (Table 1).

### 3.2. Characterization of the Generated OE Mutant Strains

#### 3.2.1. Overexpression Does Not Affect Growth in either Complete or in Minimal Liquid Media

We applied three control strains in these experiments. CPRI is the *C. maltosa LEU2* complemented derivative of the leucine auxotroph CPL2 strain, which was used to generate the mutant collection [28]. CLIB214 is a prototroph isolate and the parental strain of CPL2. We also included an mCherry fluorescent protein-expressing strain as a control OE strain (mCherry^OE^), which was generated using the same procedure. We found this pertinent because expressing an irrelevant gene (coding a “neutral” protein) in terms of virulence would imitate the same load on the expression machinery, which itself might have an effect on the physiological and virulence properties of the fungus [53].

First, we tested if the overexpression of the selected genes had any effect on the viability of the mutants. To achieve this, we cultivated the strains in YPD and YNB minimal liquid media for 24 h and compared the growth curves to that of the control strains. As a result, we found no significant differences between the growth capacity of the generated mutants compared to the three control strains (Appendix A), suggesting that the overexpression of these genes did not alter the viability of the OE mutants. Therefore, we included all the strains in the subsequent experiments to examine potential alterations in their virulence and fitness under different stress-inducing conditions, and as a control, we included the wild type CLIB214 strain unless otherwise stated.

#### 3.2.2. Overexpression of Certain Genes Affects Viability under Specific Stress Conditions

As a surrogate for the yeast’s ability to resist severe growth restrictive conditions such as elevated temperatures, different pH levels, or oxidative stressors in the host, a wide variety of the aforementioned stress conditions were applied in vitro to the OE collection on YPD solid media (Appendix A). Growth was monitored both at 30 and 37 °C, which represent the regular cultivation temperature of the fungus and the physiological body temperature of humans, respectively. The viability of the mutants was also tested at different temperatures, including 20, 25, 30, 37, and 40 °C on YPD agar plates and at 30, 37 °C on YNB and YNB + FBS solid agar. In this range of temperature, we could not detect any difference in the growth of the examined mutants in comparison to the applied controls, either on complete or minimal media (Appendix A).

Three out of the 37 overexpression mutants showed alteration in their fitness under at least one condition (Table 2). The three control strains had similar growth patterns under different stress conditions (Appendix A). Reduced growth was detected in the presence of Calcofluor white (CFW) or Congo red (CR) with CPAR2_200040^OE^ and CPAR2_302400^OE^. Susceptibility to sodium dodecyl sulfate (SDS) occurred in the case of CPAR2_109520^OE^ and CPAR2_302400^OE^, while caffeine intolerance was detected with the CPAR2_302400^OE^ mutant (Table 2). The applied stressors like CFW, CR, and caffeine have negative effects on the wall biogenesis and activate cell wall integrity (CWI) signaling via different components of the pathway. SDS is a membrane perturbing agent, and it has an effect on the integrity of the cell wall as well. Also, SDS can induce protein denaturation and cell lysis [57].

To evaluate the spot assay experiment, the following categories were assessed to compare the growth of the mutants to that of the wild type: No growth (in contrast to the wild type, the mutant did not form any colonies); strong defect (reduced growth was seen in the 10^4^–10^5^ spots, and there was usually no growth at lower cell concentrations); medium defect (reduced growth capacity was observed with the 10^3^–10^4^ spots, the 10^5^ spot showed normal growth, and, usually, the 10^2^ had no growth); slight defect (reduced growth capacity was observable at 10^2^–10^3^ spots, and, usually, the 10^4^–10^5^ spots showed normal growth); no difference (the viability of the mutant was indistinguishable from that of the control).

The fitness of the CPAR2_302400^OE^ strain decreased upon exposure to each of the cell wall–and membrane perturbing agents tested (Figure 1). The viability of this strain in the presence of CR or CFW was reduced in a concentration-dependent manner. Increasing concentration of CR resulted in medium (10, 25, 50, 75 µg/mL) to strong (75 and 100 µg/mL) growth defect. No growth or strong growth defect was observed when CFW was applied in a concentration of 50 µg/mL, while no defect was found at lower stressor concentrations. CPAR2_302400^OE^ was the only mutant that showed a growth defect in the presence of caffeine. A strong growth defect was observed when the cells were incubated in the presence of the membrane perturbing agent SDS.

Slight, medium, or strong reductions in fitness were observed when cells were exposed to CR or CFW agents during the investigation of the CPAR2_200040^OE^ mutant (Figure 1).

The CPAR2_109520^OE^ showed a medium growth defect only in the presence of SDS (Figure 1).

#### 3.2.3. The Examined OE Mutants Did Not Display Any Alterations in Morphology or Pseudohypha Formation

In contrast with *C. albicans, C. parapsilosis* is unable to form true hypha, although it is able to produce pseudohypha. The capability of reversible switching between yeast and filamentous form is an important virulence factor as it can enhance yeast surface/size and facilitate adhesion. Adhesion can lead to biofilm formation and colonization, which contribute to the intrusion of the fungus into the host cells and subsequent dissemination inside the host. It can also play a protective role for the fungus against the host immune cells and can be a tool to avoid phagocytosis [58]. Genetic elements responsible for dimorphic transition have been explored in *C. albicans* by gene deletion or expression analysis, and there have been further attempts to reveal morphological and filamentation regulatory functions in other *Candida* species [59,60,61,62,63]. Studies have concluded that the filamentary regulation pathways revealed in *C. albicans* are partially (evolutionary) conserved compared to other *Candida* species (*C. parapsilosis*, *C. tropicalis*, *C. guilliermondii*, *C. dubliniensis*) [61,64].

The screen of the OE strain collection for potential alterations in pseudophypha formation in two different liquid media at 37 °C and 5% (*v*/*v*) CO_2_ using a light microscope (qualitative analysis) as well as the Amnis^®^ FlowSight^®^ flow cytometer (quantitative analysis) revealed no detectable alterations in the mutants’ capacity to form pseudohypha (Appendix A).

#### 3.2.4. Overexpression of the *CPAR2_302400* Gene Resulted in Attenuated Biofilm Formation

Adherence and morphology switching can facilitate or strengthen biofilm formation and the virulence of a pathogenic fungal species, and *C. parapsilosis* forms biofilms on the surfaces of various medical devices, enabling nosocomial systemic infections [65]. Holland et al. examined *C. albicans* and *C. parapsilosis* biofilm-associated genes and concluded that different factors are responsible for biofilm development in these two species. While *CZF1*, *UME6*, *CPH2*, and *GZF3* are the major regulators for *C. parapsilosis* biofilm formation, in *C. albicans BRG1* and *TEC1* were found to be the central regulators of this process [28,66]. Building on the data gained from *C. parapsilosis* transcriptome and KO collection analyses, we aimed to utilize the complementary OE strategy to investigate this attribute of the fungus under biofilm-inducing conditions by monitoring their metabolic activity. We found that only CPAR2_302400^OE^ showed attenuated biofilm formation compared to the control strain (Figure 2 and Appendix A).

#### 3.2.5. Overexpression of Selected Genes Did Not Affect Tolerance to Echinocandins

Echinocandins represent the primary therapy to treat candidaemia in clinical practice; however, *C. parapsilosis* isolates show higher tolerance to these drugs compared to other *Candida* species [2,17,67]. A possible explanation for this is that *C. parapsilosis* isolates tend to accumulate point mutations in the hot spot region of antifungal resistance-related genes, causing a gain-of-function phenotype leading to increases in the tolerance to echinocandins [16]. To test how the overexpression of the selected genes affects the susceptibility of the mutants to echinocandins, the MICs of each of the strains to the three major members of this class of drugs were determined. The OE mutants displayed similar MICs to those of the control strains independently of whether anidulafungin (ANF), caspofungin (CAF), or micafungin (MIF) was applied. After 48 h of incubation, the MIC_ANF_ values of the mutants and the *C. parapsilosis* control strains were 2–4 µg/mL, both MIC_CAF_ and MIC_MIF_ values were 1–2 µg/mL, while higher sensitivity was recorded in the case of *C. krusei* (MIC_ANF_: 0.125–0.25 µg/mL; MIC_CAF_: 0.5 µg/mL; MIC_MIF_: 0.25 µg/mL). These results are in line with the literature [68,69] (Appendix A).

#### 3.2.6. Overexpression of Certain Genes Interferes with Resistance to Phagocytosis by J774.2 Cells

Phagocytes represent the primary cellular components of the innate immune system to fight against *Candida* cells reaching the tissues. Macrophages are among the first line of cellular responders that combat fungal invasion by rapidly infiltrating to the site of the infection, where they attempt to ingest and eliminate yeast cells. However, successful fungal pathogens have evolved specific mechanisms through which they are capable of interfering with macrophage responses, and some can even survive within macrophages, indicating that the capability of an intruding pathogen to avoid endocytosis and lysis by the host immune system is an important virulence trait [70,71,72,73]. To investigate if the OE strains had altered interactions with host phagocytic cells, we fluorescently labeled the OE strains with Alexa Fluor^TM^ 488 succinimidyl ester dye (Invitrogen) and co-cultured them with J774.2 murine macrophage-like cells in vitro. We monitored the phagocytic efficiency after 1 h using Amnis^®^ FlowSight^®^ flow cytometer by determining the percentages of the phagocytic activity of each sample and normalized to CLIB214 control values. We identified three strains (CPAR2_108840^OE^, CPAR2_109520^OE^, CPAR2_200040^OE^) that were more efficiently phagocytosed and two strains (CPAR2_406400^OE^, CPAR2_500180^OE^) that were less effectively phagocytosed by murine macrophages (Figure 3).

#### 3.2.7. Overexpression of Certain Genes Regulate Virulence Related Processes in *Galleria mellonella* Larvae Model

*Galleria mellonella* larvae possess similar humoral and cellular innate immune mechanisms as mammalian hosts; therefore, this insect is widely used as a substitute model for vertebrate infection experiments in order to reduce mammalian consumption. To reveal if the overexpression of the selected genes had an effect on the virulence of *C. parapsilosis* in this model, *G. mellonella* larvae were infected with the OE strains, and their survival was monitored. Shortly after the injection, the larvae became brownish due to a process called melanization. This is a common and rapid reaction of insects against any kind of foreign particles (including pathogens), whereby the hemocytes surround the foreign particle and start producing melanin via a multistep biochemical process [74]. In the case of *G. mellonella,* the melanization occurs within minutes after injection with *C. parapsilosis*. Melanization failed to occur following infection with CPAR2_302400^OE^, but there were similar survival rates to that of the control strain. However, we identified three melanin-inducing strains (CPAR2_107240^OE^, CPAR2_109520^OE^, and CPAR2_602820^OE^) that were more virulent compared to the control strain (Figure 4)

#### 3.2.8. Overexpression of the Selected Genes Does Not Affect Certain Components of the Cell Wall

Based on the results of the aforementioned experiments, certain components of the cell wall (chitin and its oligomer compounds and alpha mannan content) were further examined using specific cell wall stains. We found that none of the examined mutant strains showed an alteration compared to the control strains, neither in the distribution nor in the amount of the examined elements of the cell wall (Appendix A).

#### 3.2.9. *CPAR2_109520* and *CPAR2_302400* Differentially Alter Fungal Burden in Organs within *Mus Musculus*

We next examined the virulence of two OE mutant strains in a mouse infection model, which represents the “gold” standard of in vivo infection models. We selected CPAR2_109520^OE^ because it was more virulent than the control strain in the *G. melonella* model. We also chose CPAR2_302400^OE^, which was not significantly different from the wild type in terms of the survival of *G. mellonella*, but it did not induce melanization in the larvae. In the case of CPAR2_302400^OE^, we detected elevated fungal burdens in the brain and kidney tissues, while there were lower CFUs in the spleens and livers. CPAR2_109520^OE^ also had higher CFU levels in brain samples, while there was a reduced splenic fungal burden (Figure 5).

## 4. Discussion

The aim of this study was to generate and rigorously examine an OE mutant strain collection with the purpose of identifying virulence factors in *C. parapsilosis*. We investigated stress tolerance, morphology switch, biofilm-forming capacity, phagocytosis, and in vivo virulence in *G. mellonella* larvae and BALB/c mice. During our examinations, we found eight OE mutants with altered phenotype compared to the applied control strains. Both the CPAR2_109520^OE^ and CPAR2_302400^OE^ mutants showed an altered phenotype under three conditions, the CPAR2_200040^OE^ strain under two conditions, while the other five mutant strains under a single condition compared to the control strains (Figure 6, Appendix A). In addition, the CPAR2_109520^OE^ and CPAR2_302400^OE^ mutants were selected for murine experiments, and alterations in CFU values in different tissues were found relative to the wild type control strain. During the investigation of morphology, pseudohypha formation, and antifungal susceptibility, we detected no alterations in the OE strains compared to controls (Figure 6).

Recently, the function of three of the examined 37 genes was determined by Cillingova and co-workers [19]. The *CPAR2_100460* (*CpHBT4*), the *CPAR2_100470* (*CpHBT3*), and the *CPAR2_204840* (*CpHBT2*) genes were confirmed as hydroxybenzoate transporters. The investigators reported that the *CpHBT4* deletion mutant showed resistance to caffeine and altered sensitivity to caspofungin and fluconazole; however, we found no alteration under any tested conditions in the case of the OE parallel mutant strains. The function of 18 of the 37 genes’ orthologs has already been confirmed in *S. cerevisiae*, while 15 orthologs functions have been examined in *C. albicans* [25]. Notably, *CPAR2_200040* has no confirmed functional ortholog in either of these other yeast species.

In order to further evaluate the results, we compared the data from the OE collection analysis to the results derived from previous KO mutant library characterizations. According to Prelich, during comparison of the KO and the OE phenotype of the same gene, three phenomena can be observed: opposite (hypermorph or hypomorph) phenotypes (−/+; +/−); similar phenotypes (−/−; +/+; 0/0); no phenotype versus altered (neomorph or antimorph) properties (−/0 or 0/−; +/0 or 0/+) [36]. Amongst the studied OE mutant strains here, there are 12 [29], four [28], and three [19] (18 different genes in total) KOs available in *C. parapsilosis* that were characterized with similar methods, and importantly, they have common genetic background enabling proper comparison of the mutant pairs (Appendix A).

The ortholog gene to *CPAR2_109520* encodes a transcriptional corepressor protein, *CaTUP1*, which represses filamentous growth [75]. The CPAR2_109520^OE^ mutant displayed alterations in stress tolerance, phagocytosis, and both in vivo experiments. This OE mutant showed a medium growth defect only in the presence of SDS at 30 and 37 °C and was more avidly endocytosed by macrophages. A similar reduction in growth was also observed in a *C. parapsilosis* deletion mutant of the gene in question (with additional sensitivity to other conditions as well) [28]. The *C. albicans TUP1* null mutant showed abnormal colony morphology and increased filamentous growth compared to its originating strain [75,76,77,78]. A previous study also examining virulence in a *C. albicans TUP1* KO found reductions in adhesion, invasion, and damage with oral epithelial cells [79]. Although we also recorded reduced virulence properties of the CPAR2_109520^OE^ mutant in in vitro experiments, we found this mutant to be more virulent than the wild type *C. parapsilosis* control strain in *G. mellonella* model.

The putative ortholog of the *CPAR2_302400* gene functions as a DNA repair methyltransferase (*ScMGT1*) in *S. cerevisiae* [80,81]. The null mutant of this gene in *S. cerevisiae* showed increased growth in competitive fitness analysis [30], but there was no significant alteration in a biofilm assay [82]. A *C. albicans* null mutant also displayed normal phenotype in fitness analysis [27]; however, a *C. parapsilosis* KO showed decreased adhesive properties, but there was not any difference in its biofilm-forming capacity compared to the control strain [83]. In the current study, the *G. mellonella* model confirmed the importance of melanization against invading pathogens. The only OE strain that was coupled with the absence of melanization was CPAR2_302400^OE^, which produced a survival curve similar to that of the control strain. Although we could not detect any alteration in phagocytosis efficiency, the mutant showed severe growth defects in other important virulence properties such as stress tolerance or biofilm formation. The fitness of the CPAR2_302400^OE^ strain decreased upon exposure to each of the cell wall and membrane perturbing agents (CFW, CR, SDS, and caffeine). This mutant was the only one out of the 37 tested whose growth was reduced in the presence of caffeine and the only one that showed decreased biofilm production. Biofilm formation enables invasion while concomitantly providing protection of the pathogen from different antifungal agents and helps avoid recognition by host immune cells. *C. parapsilosis* is able to form biofilm on the surfaces of various medically used devices, thus it is a notable virulence factor [65]. This is the first report to connect the *CPAR2_302400* gene function to the biofilm regulatory network of *C. parapsilosis*, and thus to the pathogenicity of this species.

To further investigate the virulence properties of the strains, we selected two OE strains for study in a mammalian model. We chose a strain with increased virulence in the larval model, CPAR2_109520^OE^, and one that did not induce larval melanization but had virulence similar to the control strain, CPAR2_302400^OE^. The *G. mellonella* larvae and murine data suggest different invasion efficiencies of the mutants in diverse tissues or various mechanism or capacity of fungi depletion in the examined organs. Significantly higher fungal burdens were enumerated in the brains of mice infected with the CPAR2_109520^OE^ mutant, while lower CFU numbers were observed in the spleen. For CPAR2_302400^OE^, which was similar to wild type control yeast in *G. mellonella,* significantly higher CFUs were found in both the brains and kidneys of challenged mice, while lower counts were enumerated in the spleens and livers. Other comparative studies examining the correlation between the results derived from the wax moth and murine infections have also shown discordant results [84,85,86]. Nevertheless, monitoring survival in mice is not optimal in the case of many NAC species like *C. parapsilosis*, since they do not cause lethal infections in this species, while lethality can be measured in insect models [87]. Interestingly, in a mouse systemic infection model, the *C. albicans TUP1* (*CPAR2_109520* gene’s ortholog) null mutant, showed decreased virulence [78], whereas another study reported no mortality 30 days after i.v. infection with the same mutant [88]. Further, *C. albicans MGT1* (the ortholog of *CPAR2_302400*) null mutant displayed no altered phenotype in competitive experiments during i.v. infection in mice [27]. These findings underscore the imperative to explore pathogenesis studies in individual *Candida* species as determinations made with *C. albicans* cannot routinely be extrapolated to have similar behaviors in these biologically different pathogens.

The CPAR2_107240^OE^ and CPAR2_602820^OE^ mutants were also more virulent in *G. mellonella* experiments. The *CaKTR4/MNT4* (*CPAR2_107240* gene’s ortholog) gene was verified as a mannosyltranferase in *C. albicans* where it has a role in N-linked glycosylation and cell wall regeneration [89]. The null mutant of the gene showed normal growth capacity, morphology, and competitive fitness in pooled mice infection model [27]. Only the double, triple, quadruple, or quintuple deletion of the *CaMNT1-2-3-4-5* genes in different combinations (but not the single ones) resulted in mutants with altered cell wall content and elevated sensitivity to cell wall perturbing agents (CFW or SDS), indicating that *MNT* genes of *C. albicans* have redundant functions [89]. Changes in the composition of the cell wall imply alterations in host-pathogen interactions as well [89]. The *CPAR2_602820* ortholog gene (*CaFCA1*) encodes a cytosine deaminase, an enzyme of pyrimidine salvage [90,91]. The *C. albicans* null mutant was resistant to flucytosine [92]. However, there has been no information about the ortholog genes in connection with host-pathogen interactions in *C. parapsilosis*.

The CPAR2_200040^OE^ mutant displayed defects in stress tolerance and phagocytosis compared to control strains. The mutant showed slight, medium or strong reduction in fitness in the presence of CR and CFW and displayed elevated uptake by murine macrophages. The function of this gene ortholog’s function is unknown. Interestingly, the corresponding KO mutant of this strain was resistant to CFW compared to the control strain, further confirming the role of *CPAR2_200040* in cell wall assembly regulation of *C. parapsilosis* [29].

The CPAR2_108840^OE^, CPAR2_406400^OE^, and the CPAR2_500180^OE^ mutants showed altered phenotypes only in the context of phagocytosis experiments. The CPAR2_108840^OE^ mutant was more efficiently phagocytosed by macrophages. The function of *ScSPS1* (*CPAR2_108840* gene’s ortholog) is putatively a serine/threonine kinase, which is involved in spore packaging in *S. cerevisiae* [93]. Its KO parallel was sensitive to SDS in *C. parapsilosis* [29], while it displayed a normal phenotype in *C. albicans* [27]. However, the gene has not previously been linked to virulence. We also recorded lower phagocytic events with the CPAR2_406400^OE^ mutant strain. The ortholog gene, *ScRPA12,* was identified in *S. cerevisiae* as an RNA polymerase I subunit [94]. Chaillot and colleagues found that this gene has a role in cell size homeostasis in *C. albicans* [95], but it has not been associated with virulence regulation. The CPAR2_500180^OE^ mutant displayed a more virulent phenotype in phagocytosis experiments compared to the control strain. Its gene ortholog is *ScKIN3,* which encodes a nonessential serine/threonine protein kinase that has a possible role in DNA damage responses [96,97,98]. Its KO parallel was susceptible to Congo red and caffeine in *C. parapsilosis* [29] and showed increased cell size and decreased resistance to caspofungin in *C. albicans* [99,100]. In *S. cerevisiae,* its overexpression caused a reduction in vegetative growth, abnormal cellular morphology, and cell cycle [45]; however, we did not observe any of the aforementioned effects. According to the phagocytosis experiments in this study, we monitored alterations occurring at the phases of recognition and uptake. Both recognition and endocytosis primarily depend on the recognition of specific cell wall components of the pathogen. Fungal evasion of recognition occurs via masking these structures or changing morphology type by the fungi [70]. Our results suggest that differences in cell wall composition of these OE mutant strains cause the observed altered phagocytic efficiencies.

During the OE and KO phenotype comparisons, we found variations in phenotypes in the case of CPAR2_200040^OE^ when we examined CFW tolerance and in vitro host-pathogen interactions [29]. Similar phenotypes were observed with two OE mutants and their KO counterparts. The *C. parapsilosis* KO pair of the examined CPAR2_109520^OE^ mutant was sensitive to SDS. In addition to this, the CPAR2_302400^OE^ mutant strain showed decreased adhesive properties, although there was no difference in its biofilm-forming capacity compared to the control strain, while the OE mutant displayed less biofilm formation [29]. We also found absences of phenotypic alterations in OE, KO, or both parallel mutants based on the literature. The *CPAR2_108840* deletion mutant was resistant to SDS, while we did not notice such alterations in the OE strain [29]. Neither the CPAR2_500180^OE^ mutant nor its KO pair displayed any alterations under similar experimental conditions [29]. For the CPAR2_602820^OE^ mutant, we found no changes in its fitness during the spot assay analysis, while its deletion strain showed mild and strong defects in growth on minimal media at 30 and 37 °C (Appendix A). When only *S. cerevisiae* or *C. albicans* mutant collection analysis data were available, comparisons were more challenging. Additionally, it was necessary to consider whether the overexpressed genes were not unique or correctly manifest when a lack of gain-of-function or no reversed phenotype was observed [36]. In several cases, the OE and the KO phenotypes of a given gene did not complement each other. In addition to this, the applied control strains could influence the interpretation of the results. Therefore, in approaching this task, we also assessed the use of the reintegrated CPRI and an overexpressing strain mCherry^OE^ as controls in addition to the wild type CLIB214 in fitness-related experiments. We found that it was more appropriate to apply the neutral gene carrier OE strain mCherry^OE^, rather than a mutant bearing an empty cassette as a control in order to exclude any alteration in fitness derived from the integration and/or the continuous operation of the expression cassette itself (due to the presence of a constitutive promoter) [42,46,53]. As we found no alteration between the three control strains that we examined in preliminary investigations, we selected CLIB214 wild type as the control for subsequent experiments.

Although we examined diverse effects in our OE collection, several important additional areas remain to be explored if genes are examined in a targeted manner using gene deletion or overexpression strategies. These include the (1) gene dosage effect, (2) information about their interactions with other genes and (3) the subsequent transcriptomic effects, (4) off-target mutations, (5) universal behavior in various strains, and (6) the RNA level of an overexpressed gene (what we can measure) does not necessarily correlate with that of the protein it encodes [36,45,101,102,103]. Similar to our findings, previous studies in *C. albicans* did not find any OE mutants with increased fitness [42,46]. Notably, there is a substantial energy cost to a cell when a gene is duplicated or artificially overexpressed. According to Wagner, increasing protein synthesis above 10% negatively affects the health of the cell [104]. The function of the query gene also can determine the resulting phenotype, for example, periodically changing genes like cell cycle control genes [45]. So, the detected phenotype depends on the genuine function of the targeted gene. Nevertheless, our work clearly indicates that the application of the OE approach is an effective starting point to explore gene functions, especially when a prescreen (RNAseq data or known function of orthologs) for selected genes is possible. Rapid systematic analysis of the OE mutant collection (including stressors, biofilm formation, in vitro and in vivo infection models) can identify previously unknown virulence-associated genes, which can be analyzed further to reveal gene-specific functions.

In summary, this study was the first attempt to use an OE strategy to comprehensively examine gene function in *C. parapsilosis*. Significantly, the findings clearly associate *CPAR2_107240*, *CPAR2_406400*, and *CPAR2_602820* with the virulence of *C. parapsilosis.* Our results show that OE approaches are a relevant method for gene function analysis in this lethal, emerging fungus and that OE methods can define new virulence pathways in *C. parapsilosis*.

## Figures and Tables

**Figure 1 jof-07-00097-f001:**
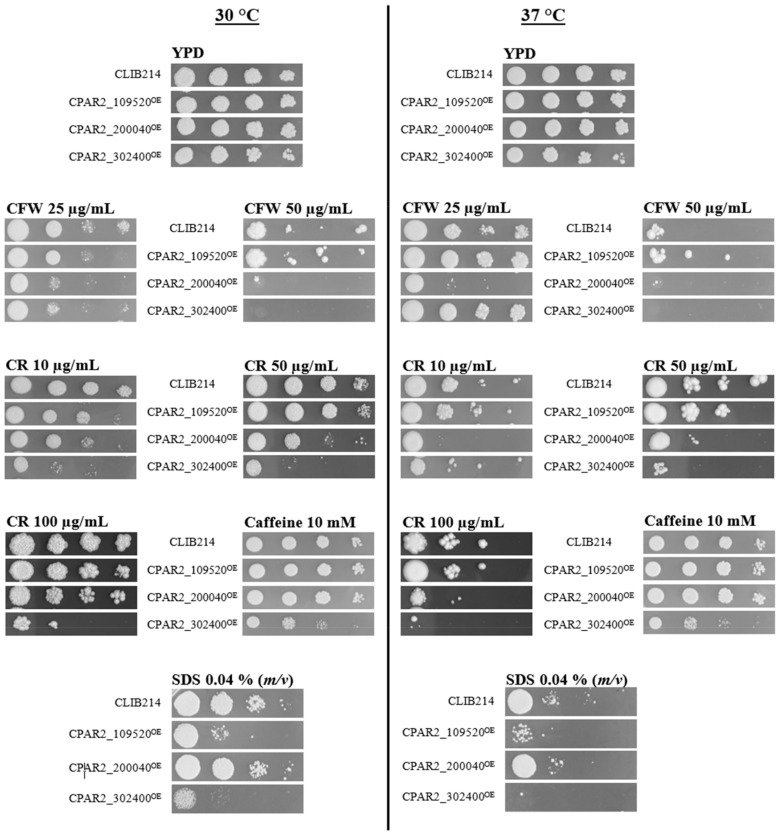
Representative pictures present CPAR2_109520^OE^, CPAR2_200040^OE^, and CPAR2_302400^OE^ strains with altered fitness compared to CLIB214 control strain in spot assay analysis. Tenfold dilutions of the cell suspensions ranging from 10^5^ cell/5 µL to 10^2^ cell/5 µL were pinned onto the surface of the solid media and kept for 48 h at 30 or 37 °C. Except for CFW (50 µg/mL) and CR (50, 100 µg/mL) at 37 °C, that was incubated for 3–4 days.

**Figure 2 jof-07-00097-f002:**
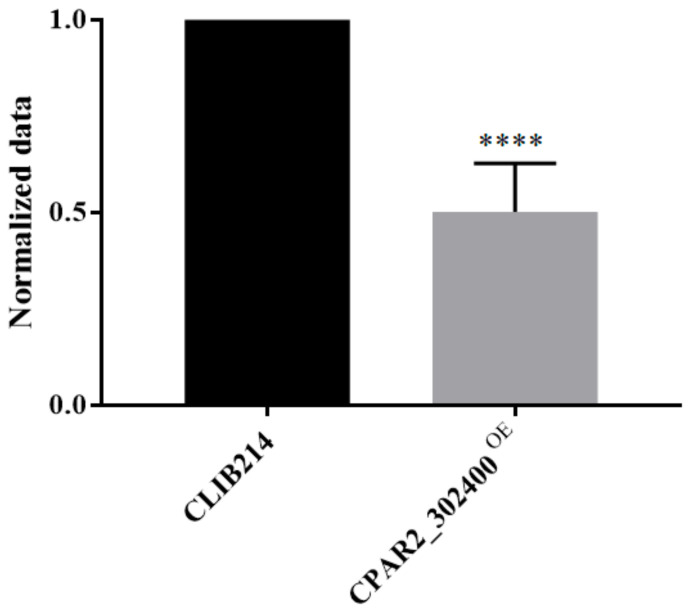
Biofilm formation analysis. OD_540_ values of the overexpression (OE) strains were normalized to the CLIB214 control strain, *n* = 3 with eight parallel samples per experiment. Statistical analysis: one-way ANOVA, Dunnett’s multiple comparisons test (**** *p* < 0.0001).

**Figure 3 jof-07-00097-f003:**
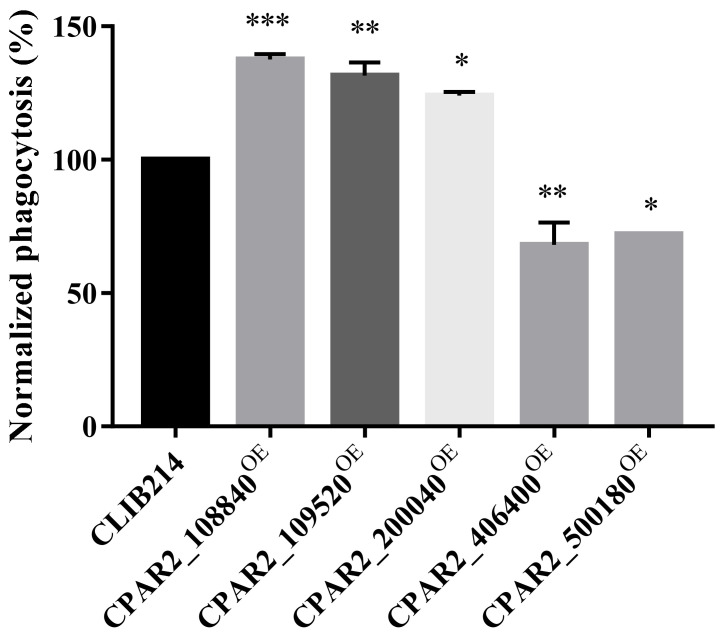
J774 phagocytosis analysis. Phagocytic activity (%) of the J774 macrophages during the co-incubation with OE strains were normalized to the values of the phagocytosed CLIB214 control strain, *n* = 3. Statistical analysis: one-way Anova, Dunnett’s multiple comparisons test (* *p* ≤ 0.05; ** *p* ≤ 0.01; *** *p* ≤ 0.001).

**Figure 4 jof-07-00097-f004:**
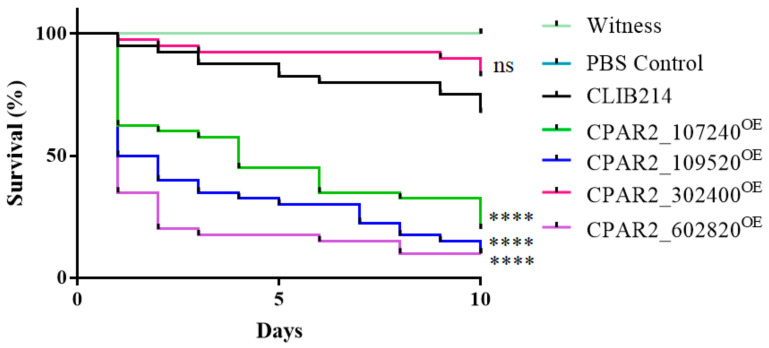
Summarized results of the in vivo *Galleria. mellonella* infection experiments, *n* = 3, separate experiments with 20 larvae per strain and experiment. Statistical analysis: Log-rank (Mantel–Cox) test (**** *p* < 0.0001; ns is not significant).

**Figure 5 jof-07-00097-f005:**
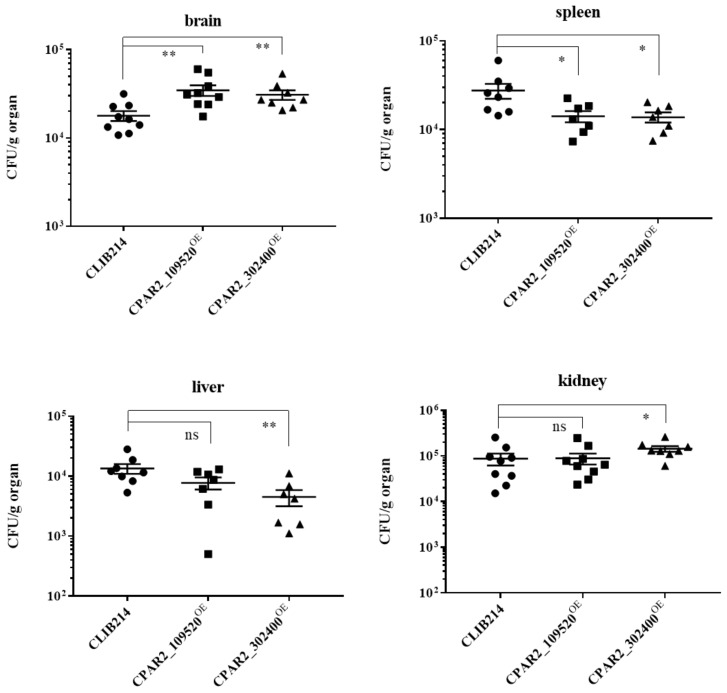
Murine infection experiments. The graphs show the fungal burden (CFU values) of brain, spleen, liver, and kidney in CFU/g organ values. Animals were infected with 2 × 10^7^ fungal cells through their lateral tail veins and sacrificed after 3 days, *n* = 2 separate experiments, with four to five animals per strain and experiment were used. For the statistical analysis Mann-Whitney tests were used (* *p* < 0.05; ** *p* < 0.01; ns: not significant).

**Figure 6 jof-07-00097-f006:**
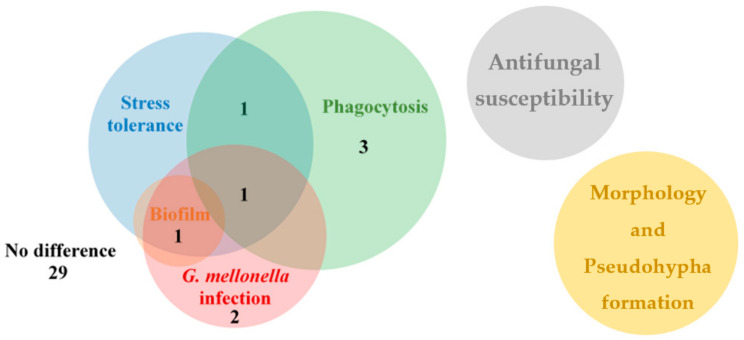
The Venn diagram shows summarized results of the OE mutant collection analysis separated into distinct experiments. The numbers display the number of mutant strains with alteration(s) in the exact experiment(s).

**Table 1 jof-07-00097-t001:** Real-time qPCR-based overexpression data (fold-change, relative normalized expression—RNE) of the created mutant strains. Data were normalized to the CLIB214 wild type strain. Relative transcription levels were determined using the housekeeping gene *CpTUB4* (*CPAR2_500510*) as an internal control. Measurements were performed in three statistical parallels. The 2^−∆∆Cq^ comparison method was used with Bio-Rad CFX Manager software. (*Cp*—*Candida parapsilosis*; *Ca*—*C. albicans*; *Sc*—*Saccharomyces cerevisiae*).

Gene Name	Ortholog Name	RNE	Gene Name	Ortholog Name	RNE
*CPAR2_100460*	*CpHBT4*	122.5 ± 17.7	*CPAR2_302400*	*ScMGT1*	424.4 ± 90.5
*CPAR2_100470*	*CpHBT3*	18.6 ± 0.9	*CPAR2_303240*	*ScPFA5*	44.7 ± 2.8
*CPAR2_100540*	*CaHAP5*	7.6 ± 0.9	*CPAR2_303730*	*ScTHR1*	66.1 ± 26
*CPAR2_104420*	*ScLTE1*	378.5 ± 22.1	*CPAR2_400270*	*ScHIT1*	31.6 ± 2.3
*CPAR2_105250*	*CaPHO85*	44.8 ± 3.5	*CPAR2_406400*	*ScRPA12*	41.7 ± 7.3
*CPAR2_107020*	*CaHHF1*	118.2 ± 11.0	*CPAR2_500180*	*ScKIN3*	675.7 ± 89.3
*CPAR2_107240*	*CaKTR4/MNT4*	70.4 ± 12.4	*CPAR2_500360*	*ScAMS1*	37.2 ± 2.6
*CPAR2_108840*	*ScSPS1*	50.5 ± 9.6	*CPAR2_501400*	*CaCWH41*	90.0 ± 11.2
*CPAR2_109520*	*CaTUP1*	19.1 ± 4.5	*CPAR2_503290*	*CaRAD53*	218.4 ± 25.9
*CPAR2_200040*	*-*	212.2 ± 17.3	*CPAR2_503760*	*ScTMA46*	7.3 ± 0.7
*CPAR2_200390*	*CaSPT3*	9.0 ± 0.6	*CPAR2_602370*	*ScRAD18*	17.3 ± 2.1
*CPAR2_201920*	*ScMNT2*	7.3 ± 1.1	*CPAR2_602820*	*CaFCA1*	28.4 ± 8.4
*CPAR2_204840*	*CpHBT2*	8.9 ± 1.8	*CPAR2_602840*	*ScTFB4*	176.7 ± 30.6
*CPAR2_205060*	*CaCDC28*	30.6 ± 5.6	*CPAR2_700550*	*ScRSC8*	4.5 ± 0.5
*CPAR2_208600*	*CaCPH1*	44.5 ± 3.6	*CPAR2_703840*	*ScSLM5*	52.5 ± 8.4
*CPAR2_209240*	*CaCDC19/PYK*	2.6 ± 0.0	*CPAR2_804030*	*ScGSC2*	134.6 ± 29.9
*CPAR2_209520*	*CaMKK2*	7.8 ± 1.3	*CPAR2_805930*	*CaTEC1*	34.5 ± 6.9
*CPAR2_300080*	*ScADK2*	134.8 ± 7.3	*CPAR2_806950*	*ScNOB1*	65.7 ± 9.9
*CPAR2_301360*	*CaPMR1*	38.9 ± 3.1			

**Table 2 jof-07-00097-t002:** The heat map shows the results of the spot assay analyses (CFW—Calcofluor white; CR—Congo red; black—no growth; red—strong defect; orange—medium defect; yellow—slight defect; grey—no difference).

Stressor	Caffeine (10 mM)		CFW (25 µg/mL)		CFW (50 µg/mL)		CR (10 µg/mL)		CR (25 µg/mL)		CR (50 µg/mL)		CR (75 µg/mL)		CR (100 µg/mL)		SDS (0.04% *m*/*v*)	
T (°C)	30	37	30	37	30	37	30	37	30	37	30	37	30	37	30	37	30	37
CPAR2_109520^OE^ (*CpTUP1*)																		
CPAR2_200040^OE^																		
CPAR2_302400^OE^ (*CpMGT1*)																		

## Data Availability

The data presented in this study are available in “A *Candida parapsilosis* Overexpression Collection Reveals Genes Required for Pathogenesis”.

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
