# Peer review of "A *Candida parapsilosis* Overexpression Collection Reveals Genes Required for Pathogenesis"

_jof, 2021, doi:10.3390/jof7020097_

Round 1

Reviewer 1 Report

The authors explore virulence-related genes by characterizing a collection of gene overexpressing strains in C. parapsilosis. This study is the first to use this strategy in this species. The engineered strains are characterized regarding their fitness in standard and stress conditions, morphology, biofilm formation, phagocytosis and virulence in two different models, Galleria mellonella larvae and mouse.  The authors identify some strains with altered phenotypes regarding the wild type.

Points to address:

1. The title could be improved: virulence associated genes are, by definition, required for pathogenesis; state what is being overexpressed (Gene overexpression collection)

2. Abstract: “This work identified CPAR2_107240, CPAR2_406400, and CPAR2_602820 genes as contributors to C. parapsilosis virulence by regulating functions as cell wall assembly and maintenance and biofilm formation.”

Through the analysis of the manuscript, I could only conclude that these genes contribute to C. parapsilosis virulence in larvae (Figure 5) and that CPAR2_406400 overexpressing strain is less phagocytosed by murine macrophages (Figure 4). Do the authors have any figure to show that the strains overexpressing these genes have an altered phenotype regarding cell wall biogenesis or biofilm formation?

3. Overall I believe that the manuscript would benefit if more information was given in the results chapter regarding on how and why the experiments were done. It would be also useful to include more detailled information regarding on how the genes were chosen for the overexpression analysis.

4. Table 1: data regarding SD (standard deviation of the three replicates) should be presented

5. Table 2 does not show which strain corresponds to each column. The quality of the image needs to be improved.

6. Figure 1 is a little confusing. It may be beneficial to select only low, medium and high concentration of the stressing agent (example: CR 10, 50 and 100 µg/ml), removing the number of cells in each spot (as it is the same for all, it can be stated at the figure legend), and/or showing the results of each condition in the three strains side by side. It would be useful to include the spots for all stressors (CR, CFW, Caffeine and SDS) for each strain, even if the strain is not sensitive to it. Control spots in YPD medium (without stressing agent) is missing. This is important to rule out dilution and fitness issues. Strains sensitivity to CR do not seem to increase in a concentration dependent manner, this is particularly apparent in CPAR2_200040OE strain. Where the pictures taken and processed likewise?

7. Line 303-307: data not shown regarding CFW - please consider, as mentioned before, adding to the figure the result for CFW 25 μg/ml, even if the strain is not sensitive.

8. Line 311-313: the temperature dependency is also observed in the wild type. It seems rather specific to the species rather than to the CPAR2_200040OE strain.

9. Figure 2: it would be useful to show the biofilm formation results for all the strains tested in the supplementary data.

10. Section 3.2.5: the data should be shown, at least in supplementary material, as it was done for pseudohyphae analysis.

11. It would be useful to include in the manuscript (perhaps replacing Figure 7) a table summarizing the obtained results and a comparison with data (phenotypes in similar assays) from related KO mutants from C. parapsilosis (if available) or from orthologs. In fact, it would be interesting if in this work, new data on KO mutants from C. parapsilosis was shown, so it could be directly compared to the gene overexpressing strains.

12. Lines 425-426: what does it mean and how is it measured “larval melanization”? Less virulent but similar survival?

Minor issues:

13. No space should be inserted between a number and the percentage symbol (e.g. 25%).

14. line 43: …albicans…; line 46: …pseudohyphae…; line 72: …such as biofilm…; line 76: Although extensive…; lines 133, 203 and 251: wild type; line 125 …Southern blot…10 µg…; line 142: …growth kinetics…; lines 178 and 249: replace comma for point; line 204: “cells” missing; line 281: (additionally) 30 and 37 °C has already been mentioned; line 518: …fluconazole…; line 556: …to connect…

15. Please consider reducing the number of citations in the first paragraph of the introduction.

16. Is it necessary to include the “Applied primers” section (2.2.) in Materials and Methods?

17. lines 293-298: unclear

18. Legend of Figure 4: macrophages have a phagocytic activity, not Cp strains

19. There are some format errors in the references (5, 6, 13, 24, 57, 58 and 88).

20. Table S3 appears as Table S1 in the title.

21. The legend of table 1 should include the meaning of Cp, Ca and Sc.

Author Response

Reviewer
We would like to thank our Reviewer for the extensive and precise evaluation of our manuscript and for emphasizing its strengths and pointing out its weaknesses. We really appreciate the detailed questions, critiques and remarks attached.

1. The title could be improved: virulence associated genes are, by definition, required for pathogenesis; state what is being overexpressed (Gene overexpression collection)

We agree with the suggested alterations in the title. Indeed, the simultaneous usage of the terms „virulence associated genes” and „pathogenesis” is self-explanatory.

2. Abstract: “This work identified CPAR2_107240, CPAR2_406400, and CPAR2_602820 genes as contributors to C. parapsilosis virulence by regulating functions as cell wall assembly and maintenance and biofilm formation.”
Through the analysis of the manuscript, I could only conclude that these genes
contribute to C. parapsilosis virulence in larvae (Figure 5) and that CPAR2_406400
overexpressing strain is less phagocytosed by murine macrophages (Figure 4). Do the authors have any figure to show that the strains overexpressing these genes have an altered phenotype regarding cell wall biogenesis or biofilm formation?

We agree with our Reviewer, this sentence was mistakenly stated, we interpreted our results in an incorrect way. Here, we wanted to emphasize that our findings were the first report which correlated the following 3 genes (CPAR2_107240, CPAR2_406400, and CPAR2_602820) to the pathomechanism of C. parapsilosis through their altered phenotype in the examined hostpathogen interactions. The orthologs of CPAR2_107240, CPAR2_406400 and CPAR2_602820 in C. albicans are associated with cell wall maintenance, adhesion/hyphae or biofilm formation and adherence to polystyrene surface, respectively [1,2]. To investigate the
role of these genes in the related mechanisms the content of certain cell wall components and the adhesion properties of the mutants were tested. Total chitin (calcofluor white), alpha mannan (Concavalin A) and chitin oligomer compounds (Wheat Germ Agglutinin) were dyed, but we couldn’t notice any alterations in either microscopic or flow cytometry measurements. A representative supplementary figure of the result of the cell wall staining is attached to the
revised version of the manuscript. We also attached the results regarding the biofilm formation as suggested by our Reviewer in “Point 9”. We also completed this section with other 5 genes (CPAR2_108840, CPAR2_109520, CPAR2_200040, CPAR2_302400, CPAR2_500180) whose OE caused altered phenotypes in
our experiments, however our study was not the first one which examined or reported them or their orthologs as virulence related genes in C. parapsilosis or in C. albicans. It has already been established that the lack of genes CPAR2_108840, CPAR2_109520, CPAR2_200040, CPAR2_302400 and CPAR2_500180 results in altered phenotype under specific circumstances, out of which only CPAR2_109520, CPAR2_200040 and CPAR2_500180 were linked to virulence so far.

3. Overall I believe that the manuscript would benefit if more information was given in the results chapter regarding on how and why the experiments were done. It would be also useful to include more detailed information regarding on how the genes were chosen for the overexpression analysis.

We thank the remark of our Reviewer. We revised the “Results” section of our manuscript accordingly. 18 genes were chosen for the OE analysis based on previously published data regarding KO mutants [3-5] and 19 genes were selected based on their ortholog functions. A short description is added to the first paragraph of the results section.

4. Table 1: data regarding SD (standard deviation of the three replicates) should be presented

Indeed, it is more adequate to include SDs as well. Overexpression data is now presented as average and SD.

5. Table 2 does not show which strain corresponds to each column. The quality of the image needs to be improved.

Indeed, unfortunately the titles of this figure were slipped away. We corrected and edited it.

6. Figure 1 is a little confusing. It may be beneficial to select only low, medium and high concentration of the stressing agent (example: CR 10, 50 and 100 μg/ml), removing the number of cells in each spot (as it is the same for all, it can be stated at the figure legend), and/or showing the results of each condition in the three strains side by side.

It would be useful to include the spots for all stressors (CR, CFW, Caffeine and SDS) for each strain, even if the strain is not sensitive to it. Control spots in YPD medium (without stressing agent) is missing. This is important to rule out dilution and fitness issues. Strains sensitivity to CR do not seem to increase in a concentration dependent manner this is particularly apparent in CPAR2_200040OE strain. Where the pictures taken and processed likewise?
We considered the suggestions of our Reviewer and changed Figure 1 accordingly. We attached figures of the 3 mutant strains (CPAR2_109520OE, CPAR2_200040OE, CPAR2_302400OE) in spot assay analysis in the presence of CR 10-50-100 μg/ml, CFW 25-50 μg/ml, SDS (0.04% w/V) and caffeine (10 mM) and the YPD control at 30 and 37 °C as well.
Regarding the notes on concentration dependency, we stated and observed this phenomenon only in the case of the CPAR2_302400OE mutant strain in the presence of CR and CFW agents. Increasing concentration of CR resulted in medium (10, 25, 50, 75 μg/ml) to strong (75, 100 μg/ml) growth defect. No growth or strong growth defect was observed when CFW was applied
in a concentration of 50 μg/ml, while no defect was found at lower stressor concentrations. The pictures were taken and processed in the same way.

7. Line 303-307: data not shown regarding CFW - please consider, as mentioned before, adding to the figure the result for CFW 25 μg/ml, even if the strain is not sensitive.

We revised the table as suggested by our Reviewer.

8. Line 311-313: the temperature dependency is also observed in the wild type. It seems rather specific to the species rather than to the CPAR2_200040OE strain.

We thank our Reviewer for this remark. We always compared the viability of the mutants to the respective control strain and condition. Indeed, we apologize not to include the YPD control plates in Figure 1. We did so as our Reviewer requested. These indicate that the growth of both the CPAR2_200040OE strain and the wild type does not show any alterations on YPD at the two temperatures, but in the presence of Congo red a reduction in growth was visible at 37 °C, compared to growth at 30 °C. Our Reviewer is right regarding the growth of the control
strain at 37 °C (CR), its viability is decreased obviously as well. The phenotype of the mutant and also that of the control strain is affected by both the stressors (heat and CR) making it difficult to conclude which causes the reduced growth and to what extent. We agree with our Reviewer, at this point such a statement cannot be made, so we removed it from the manuscript. Even if this phenotype exists, a spot assay is not the most appropriate technique to detect it.

9. Figure 2: it would be useful to show the biofilm formation results for all the strains tested in the supplementary data.

We attached the figure containing the result of the biofilm assay (Supplementary Figure S8).

10. Section 3.2.5: the data should be shown, at least in supplementary material, as it was done for pseudohyphae analysis.
We accepted the recommendation and attached the table containing the results of the antifungal susceptibility analysis as Supplementary Table S5.

11. It would be useful to include in the manuscript (perhaps replacing Figure 7) a table summarizing the obtained results and a comparison with data (phenotypes in similar assays) from related KO mutants from C. parapsilosis (if available) or from orthologs. In fact, it would be interesting if in this work, new data on KO mutants from C. parapsilosis was shown, so it could be directly compared to the gene overexpressing strains.

We agree with the suggestion of our Reviewer. A table which summarizes all the data from the recent and previous studies from KO or OE mutants from C. parapsilosis would provide an insight on the virulence associated genes of C. parapsilosis and related phenotypes. We generated the table as recommended (Supplementary Table S6).

12. Lines 425-426: what does it mean and how is it measured “larval melanization”?
Less virulent but similar survival?

We apologize for this overstatement and for not providing a proper definition of melanization. Indeed, a non-significant difference between the survival curves can not be stated as attenuated virulence. A short description was added into the main text: “Melanization is a common and rapid reaction of insects against any kind of foreign particles (including pathogens), when the hemocytes surround the foreign particle and start producing melanin via a multistep biochemical process. In the case of G. mellonella the melanization occurs within minutes after injection with C. parapsilosis. However, we found that upon infection with CPAR2_302400OE the melanization did not occur, that was coupled with similar survival rates
to that of the control strain.”

Minor issues:
13. No space should be inserted between a number and the percentage symbol (e.g. 25%).

We corrected all the related data as requested.

14. line 43: …albicans…; line 46: …pseudohyphae…; line 72: …such as biofilm…; line
76: Although extensive…; lines 133, 203 and 251: wild type; line 125 …Southern blot…10
μg…; line 142: …growth kinetics…; lines 178 and 249: replace comma for point; line
204: “cells” missing; line 281: (additionally) 30 and 37 °C has already been mentioned;
line 518: …fluconazole…; line 556: …to connect…

We thank our Reviewer for highlighting our typos and grammar mistakes, they were corrected as requested.

15. Please consider reducing the number of citations in the first paragraph of the
introduction.

We accept the comment of our Reviewer and reduced the number of citations.

16. Is it necessary to include the “Applied primers” section (2.2.) in Materials and
Methods?

We revised and included the 2.2. section to the 2.3. (Construction of the overexpression cassettes).

17. lines 293-298: unclear

We agree, this section is indeed confusing. We clarified the description of the evaluation of the spot assay analysis in Materials and Methods section.

18. Legend of Figure 4: macrophages have a phagocytic activity, not Cp strains

We apologize for this misphrasing. We rewrote as requested.

19. There are some format errors in the references (5, 6, 13, 24, 57, 58 and 88).

We re-checked and corrected the formatting of the references.

20. Table S3 appears as Table S1 in the title.

We revised as requested.

21. The legend of table 1 should include the meaning of Cp, Ca and Sc.

Indeed, we added what these abbreviations mean

References
1. Marchais, V.; Kempf, M.; Licznar, P.; Lefrançois, C.; Bouchara, J.-P.; Robert, R.; Cottin, J. DNA array analysis
of Candida albicans gene expression in response to adherence to polystyrene. FEMS Microbiol. Lett. 2005,
245, 25–32, doi:https://doi.org/10.1016/j.femsle.2005.02.014.Pfaller, M.A.; Diekema, D.J. Epidemiology of
Invasive Candidiasis: a Persistent Public Health Problem. Clin. Microbiol. Rev. 2007, 20, 133–163,
doi:10.1128/CMR.00029-06.
2. Harcus, D.; Nantel, A.; Marcil, A.; Rigby, T.; Whiteway, M. Transcription Profiling of Cyclic AMP Signaling in
Candida albicans. Mol. Biol. Cell 2004, 15, 4490–4499, doi:10.1091/mbc.e04-02-0144.
3. Cillingová, A.; Zeman, I.; Tóth, R.; Neboháčová, M.; Dunčková, I.; Hölcová, M.; Jakúbková, M.; Gérecová, G.;
Pryszcz, L.P.; Tomáška, Ľ.; et al. Eukaryotic transporters for hydroxyderivatives of benzoic acid. Sci. Rep.
2017, 7, 8998, doi:10.1038/s41598-017-09408-6.
4. Holland, L.M.; Schröder, M.S.; Turner, S.A.; Taff, H.; Andes, D.; Grózer, Z.; Gácser, A.; Ames, L.; Haynes, K.;
Higgins, D.G.; et al. Comparative Phenotypic Analysis of the Major Fungal Pathogens Candida parapsilosis
and Candida albicans. PLOS Pathog. 2014, 10, e1004365.
5. Tóth, R.; Cabral, V.; Thuer, E.; Bohner, F.; Németh, T.; Papp, C.; Nimrichter, L.; Molnár, G.; Vágvölgyi, C.;
Gabaldón, T.; et al. Investigation of Candida parapsilosis virulence regulatory factors during host-pathogen
interaction. Sci. Rep. 2018, 8, 1346, doi:10.1038/s41598-018-19453-4.

Reviewer 2 Report

In this manuscript, Pál et al. use an overexpression (OE) approach to investigate the function of several genes involved in the pathogenesis of C. parapsilosis. These genes were selected due to either their different transcriptional expression patterns after co-incubation with THP-1 human monocytes or the known virulence of related orthologs in C. albicans or S.cereviseae. It is worth mentioning that it is the first time that an OE strategy is used to examine gene function in C. parapsilopsis. The results provided in this work not only shed light on the virulence properties of some of the genes analyzed, but also highlight the utility of OE methods to study the role of certain genes in virulence. The manuscript is mostly clear and very informative. The outcomes are properly discussed providing novel insights in the virulence of C. parapsilopsis.

To be considered:

Since the authors are analyzing so many genes, it is difficult to follow the descriptions provided in the discussion. I think that a table following the Prelich comparison (opposite phenotype, similar phenotype or no phenotype vs altered) and including bullets of the main phenotypes detected per strain would be useful.

Issues:

Table 1. "Measurements were performed in three statistical parallels" Do authors want to say biological replicates?

Table 2. It is not possible to see what strain is related to each phenotype.

Line 389. Despite OE mutants and control strains having similar MICs, it is important to at least provide the MIC of the controls. It is informative and they could be used as a reference.

Fig. 4. N=1 per strain, in 2-3 experiments. Experiment must be done at least 3 times (three biological replicates). Statistical analysis needs at least three replicates to be performed.

Line 419. Galleria mellonella is not in italics.

Line 546. Galleria mellonella or G. mellonella

Line 596-597: “further confirming the role of CPAR2_200040 in cell wall assembly regulation [41], further confirming the role of CPAR2_200040 in cell wall assembly regulation of C. parapsilosis” Text is repeated.

Author Response

We would like to thank our Reviewer for carefully reading our manuscript. We appreciate the supportive comments and remarks.

Since the authors are analyzing so many genes, it is difficult to follow the descriptions provided in the discussion. I think that a table following the Prelich comparison (opposite phenotype, similar phenotype or no phenotype vs altered) and including bullets of the main phenotypes detected per strain would be useful.

We accepted the suggestion and we also think that such a table which summarize all data stem from our and other’s studies would be very useful and can facilitate the interpretation of this study. We attached the table as requested (Supplementary Table S6).

Issues:

Table 1. "Measurements were performed in three statistical parallels" Do authors want to say biological replicates?

We thank our Reviewer for the remark. In this experiment we inoculated and cultivated each strain independently in three parallels. The RNA extraction was performed from the pool of these suspensions mixed in an even ratio. We also revised this in the materials and methods section.

Table 2. It is not possible to see what strain is related to each phenotype.

Unfortunately, the titles of this figure were slipped away. We corrected and edited it as requested.

Line 389. Despite OE mutants and control strains having similar MICs, it is important to at least provide the MIC of the controls. It is informative and they could be used as a reference.

We agreed with the remarks. We included these data into Supplementary Table S5.

Fig. 4. N=1 per strain, in 2-3 experiments. Experiment must be done at least 3 times (three biological replicates). Statistical analysis needs at least three replicates to be performed.

We thank our Reviewer for pointing out this issue. This information was mistakenly stated. We revised it in the figure legend. In the case of the indicated mutant strains we repeated the experiments 3 times. Regarding the strains when alterations compared to the wild type could not have been observed after 2 experiments the 3rd repetitions were not carried out.

Line 419. Galleria mellonella is not in italics.

Line 546. Galleria mellonella or G. mellonella

Line 596-597: “further confirming the role of CPAR2_200040 in cell wall assembly regulation [41], further confirming the role of CPAR2_200040 in cell wall assembly regulation of C. parapsilosis” Text is repeated.

We thank our Reviewer for highlighting our mistakes regarding grammar and style. Indicated sections were corrected.

Reviewer 3 Report

The manuscript by Pál and coworkers entitled “Overexpression collection in Candida parapsilosis reveals virulence associated genes required for pathogenesis” describes the generation of an overexpression mutant collection as a strategy to study virulence in C. parapsilosis. Through the characterization of the obtained mutant in in vitro and in vivo experiments, the authors provide evidence that 8 genes may contribute to C. parapsilosis virulence, by regulating cell wall assembly and integrity and biofilm formation.

The manuscript presents results that could be of interest to the Journal of Fungi readers, however it would benefit from a revision to improve the quality of the information provided.

My comments are outlined below.

The paper is well structured, and the rationale clearly stated. Methods are robust and the results are appropriately described. I only have a few suggestions to improve the quality of the manuscript.

  • In the Introduction section of the paper, the Authors describe conventional methods to study gene function in Candida, such as gene knock outs and the observation of the resulting phenotype. They also correctly point out several practical hurdles that this strategy presents. They state that overexpression studies can circumvent these problems and address the issue. It is somewhat anachronistic not to mention that gene editing has also been used to overcome drawbacks and limitations of conventional deletion techniques. Indeed, the development of the CRISPR/Cas9 based technology has revolutionized gene function studies in Candida research, allowing to rapidly generate knock outs and knock ins mutants and to simultaneously inactivate multiple members of gene families. I think this information and relative literature should be provided, especially considering that the episomal CRISPR/Cas9 system has been optimized in C. parapsilosis (Lombardi et al.,Sci Rep. 2017;7(1):8051; Lombardi  et al., mSphere. 2019;4(2):e00125-19) and used for the efficient gene editing in the C. parapsilosis species complex (Morio et al., J Antimicrob Chemother. 2019;74(8):2230-2238. Zoppo et al. Future Microbiol. 2019;14:1383-1396; Zoppo et al., 2020 Microbiol Res. 2020;231:126351).
  • Limitation(s) of the overexpression mutant generation strategy should also be clearly stated in a paragraph of the Discussion section of the manuscript. For instance, in this study the collection was generated in an auxotrophic, laboratory strain, and this may not reflect genetic variability observed among circulating clinical isolates.

Minor comments

Although generally well written, a few typos are present in the manuscript, please check the English carefully

Line 43, C. albicans is misspelled (C. albcians)

Line 202, Paragraph  2.6.8. “In vivo mice infection model” please, change into “In vivo mouse infection model” or “murine model”

Line 498 “During the investigation of morphology, pseudohyphae formation…” please, change into “During the investigation of morphology, pseudohypha formation...”  The same applies to Figure 7 (pseudophypa formation)

In supplementary Figure3 “Figure S3: Results of the growth kinetic measurements in complete media”, it is not clear what medium each curve refers to. The same applies to Figure S4. In this particular case, one or two strains seem to have a slower growth rate, but it is very difficult to understand which medium corresponds to the image (#4-6). Is this difference in growth rate among the strains considered trivial?

Line 512: The Authors state: “The investigators reported that the CpHBT4 deletion mutant showed resistance to caffeine and altered sensitivity to caspofungin and fuconazole; however, we found no alteration under any tested conditions in the case of the OE parallel mutant strains.” Was the deletion performed in the same genetic background as the one used in this study?  Could this account for a different phenotype?

In supplementary Figure 7 (please change “pseudohyphae analysis” into “pseudophypa analysis”), it is not clear which of the white bars serves as calibration bar for micrographs; please modify this accordingly.

Author Response

We are grateful to our Reviewer for thoroughly reading our manuscript, and we appreciate the remarks and the suggestions on how to improve its quality.

In the Introduction section of the paper, the Authors describe conventional methods to study gene function in Candida, such as gene knock outs and the observation of the resulting phenotype. They also correctly point out several practical hurdles that this strategy presents. They state that overexpression studies can circumvent these problems and address the issue. It is somewhat anachronistic not to mention that gene editing has also been used to overcome drawbacks and limitations of conventional deletion techniques. Indeed, the development of the CRISPR/Cas9 based technology has revolutionized gene function studies in Candida research, allowing to rapidly generate knock outs and knock ins mutants and to simultaneously inactivate multiple members of gene families. I think this information and relative literature should be provided, especially considering that the episomal CRISPR/Cas9 system has been optimized in C. parapsilosis (Lombardi et al.,Sci Rep. 2017;7(1):8051; Lombardi et al., mSphere. 2019;4(2):e00125-19) and used for the efficient gene editing in the C. parapsilosis species complex (Morio et al., J Antimicrob Chemother. 2019;74(8):2230-2238. Zoppo et al. Future Microbiol. 2019;14:1383-1396; Zoppo et al., 2020 Microbiol Res. 2020;231:126351).

We thank our Reviewer for this comment and recommendation. We supplemented the “Introduction” section with the recently developed CRISPR/Cas9 method for gene editing in C. parapsilosis and related species. When we started generating the OE mutants the CRISPR/Cas9 technology was not available in C. parapsilosis. Consequently, it is true that the Gateway strategy itself is not the most up-to-date cloning strategy nowadays, we agree, however our goal was to apply overexpression as a strategy for gene function analysis in C. parapsilosis, rather than the way how it is achieved. Addition to this, Németh and co-workers presented that the Gateway cloning approach combined with the appropriate integrative constructs can provide a reliable and constitutive expression of the given gene in question [1].

Limitation(s) of the overexpression mutant generation strategy should also be clearly stated in a paragraph of the Discussion section of the manuscript. For instance, in this study the collection was generated in an auxotrophic, laboratory strain, and this may not reflect genetic variability observed among circulating clinical isolates.

We are grateful to our Reviewer for pointing out this issue in our manuscript. A more detailed description is now provided in the “Discussion” part on the drawbacks and potential bottlenecks of the OE approach. We indeed applied an auxotrophic, laboratory strain to generate the mutant collection, however Németh and colleagues have developed a Gateway construction utilizing a dominant selectable marker (nourseothricin acetyltransferase) as well allowing us to generate overexpression mutants even in clinical isolates of C. parapsilosis [1].

Minor comments

Although generally well written, a few typos are present in the manuscript, please check the English carefully.

Line 43, C. albicans is misspelled (C. albcians)

Line 202, Paragraph  2.6.8. “In vivo mice infection model” please, change into “In vivo mouse infection model” or “murine model”

Line 498 “During the investigation of morphology, pseudohyphae formation…” please, change into “During the investigation of morphology, pseudohypha formation...” The same applies to Figure 7 (pseudophypa formation)

We thank our Reviewer the comments regarding our grammar mistakes and typos. All the sections in question were corrected.

In supplementary Figure3 “Figure S3: Results of the growth kinetic measurements in complete media”, it is not clear what medium each curve refers to. The same applies to Figure S4. In this particular case, one or two strains seem to have a slower growth rate, but it is very difficult to understand which medium corresponds to the image (#4-6). Is this difference in growth rate among the strains considered trivial?

In the Figure S3 all of the data related to YPD media, while in the case of the Figure S4 all data related to minimal media. In this latter case, some strains indeed seemed to have a reduced growth rate compared to others, however it was a general phenomenon in most of the cases as a reaction to restricted conditions. In addition to this, we also examined the growth capacity of the mutant strains on solid minimal media to exclude mutants with growth defects and we couldn’t detect any alteration compared to the wild type. According to this we considered all mutant strains having the same growth capacity as the wild type on minimal media.

Line 512: The Authors state: “The investigators reported that the CpHBT4 deletion mutant showed resistance to caffeine and altered sensitivity to caspofungin and fuconazole; however, we found no alteration under any tested conditions in the case of the OE parallel mutant strains.” Was the deletion performed in the same genetic background as the one used in this study?  Could this account for a different phenotype?

In the case of the referred C. parapsilosis KO mutants the background was a histidine/leucine double auxotroph strain (CPL2H1) [2,3,4]. To generate KO mutants HIS1 allele of Candida dubliniensis (CdHIS1) and LEU2 allele of Candida maltosa (CmLEU2) are used to complement the auxotrophy of the strain in question. In our study we applied a leucine auxotroph strain (CPL2), which is the derivative of CPL2H1 gained by complementing its histidine auxotrophy by the same CdHIS1 allele used to generate the KOs. As both KO and OE mutants utilize the very same auxotrophic selectable markers we consider our OE and the parallel KO mutant strains comparable. The Discussion section was supplemented with a short comment on the genetic background of the OE and KO strains of C. parapsilosis.

In supplementary Figure 7 (please change “pseudohyphae analysis” into “pseudophypa analysis”), it is not clear which of the white bars serves as calibration bar for micrographs; please modify this accordingly.

We thank our Reviewer for the comment, we revised the figure in question.

References

  1. Németh, T.; Papp, C.; Vagvolgyi, C.; Chakraborty, T.; Gacser, A. Identification and Characterization of a Neutral Locus for Knock-in Purposes in parapsilosis. Front. Microbiol. 2020, 11, 1194.
  2. Cillingová, A.; Zeman, I.; Tóth, R.; Neboháčová, M.; Dunčková, I.; Hölcová, M.; Jakúbková, M.; Gérecová, G.; Pryszcz, L.P.; Tomáška, Ľ.; et al. Eukaryotic transporters for hydroxyderivatives of benzoic acid. Rep. 2017, 7, 8998, doi:10.1038/s41598-017-09408-6.
  3. Holland, L.M.; Schröder, M.S.; Turner, S.A.; Taff, H.; Andes, D.; Grózer, Z.; Gácser, A.; Ames, L.; Haynes, K.; Higgins, D.G.; et al. Comparative Phenotypic Analysis of the Major Fungal Pathogens Candida parapsilosis and Candida albicans. PLOS Pathog. 2014, 10, e1004365.
  4. Tóth, R.; Cabral, V.; Thuer, E.; Bohner, F.; Németh, T.; Papp, C.; Nimrichter, L.; Molnár, G.; Vágvölgyi, C.; Gabaldón, T.; et al. Investigation of Candida parapsilosis virulence regulatory factors during host-pathogen interaction. Rep. 2018, 8, 1346, doi:10.1038/s41598-018-19453-4.

Round 2

Reviewer 1 Report

I appreciate all the effort made by the authors to improve the quality of the manuscript.

Minor issues for final proof-reading:

  1. Line 45: pseudohypha
  2. Line 66: CRISPR/Cas9
  3. Chapter “Overexpression of the selected genes does not affect certain components of the cell wall”: the number is not correct (3.2.3) or it is not in the right place. Indeed, it may be better to place this chapter after 3.2.2 and leave the in vivo models together.